## Article

# Mobile Broadband Performance Evaluation: Analysis of National Reports

Yalew Zelalem Jembre [1], Woon-young Jung [2], Muhammad Attique [3], Rajib Paul [4] and Beomjoon Kim [1,*]

1 Department of Electronic Engineering, Keimyung University, Daegu 42601, Korea; zizutg@kmu.ac.kr
2 Department of ICT Infrastructure & Platform, National Information Society Agency (NIA), Daegu 41068, Korea; wyjung@nia.or.kr
3 Department of Software, Sejong University, Seoul 05006, Korea; attique@sejong.ac.kr
4 Department of Software and Computer Engineering, Ajou University, Suwon 16499, Korea; rajib@ajou.ac.kr
* Correspondence: bkim@kmu.ac.kr

**Abstract:** Five decades have passed since the first bit was transmitted over the internet. Although the internet has improved our lives and led to the digital economy, currently only 51% of the world's population have access to it. Currently, consumers mostly access the internet via mobile broadband, 2G, 3G, and 4G services. Regulatory bodies such as the Federal Communications Commission (FCC) of the US are responsible for ensuring that consumers receive an adequate service from Mobile Network Operators (MNOs). Usually, regulators evaluate the performance of each MNO in terms of service quality yearly and publish a report. To evaluate performance, metrics such as coverage, download/upload speed, and the number of subscribers can be used. However, the evaluation process and the metrics used by each regulatory body are inconsistent, and this makes it hard to determine which nations are providing adequate services to their citizens. Furthermore, it is not clear as to which performance evaluation is the right path. In this case study, we analyzed the reports released from eight nations (United States of America, United Kingdom, France, South Korea, Japan, Singapore, and Australia) as of the year 2020. We then point out the advantages and the drawbacks of the current evaluation process and metrics. Furthermore, a discussion on why the current methods are not sufficient to evaluate 5G services is presented. Our findings indicate that there is a great need for a unified metric and that this process becomes more complex with the rollout of 5G.

**Keywords:** performance evaluation; mobile broadband; network operators; regulators; metrics

## 1. Introduction

The internet has come a long way since its inception back in 1969 [1]. It has grown exponentially in itself and also led to many breakthroughs in the field of social as well as scientific studies. The year of 2019 was monumental for the internet. It marked the 50th year of the very first transmission of data; these days, more than 74 thousand Gigabytes of data are sent per second [2]. It is also the first time that more than half of the world became part of the global digital economy by accessing the internet, that is, about 51% or 3.9 billion individuals [3]. In addition, it was also the 30th year of the World Wide Web (WWW) web browser and the 25th year of the first e-commerce exchange.

However, we have a long way to go to reach the target of connecting 75% of the world to the internet, which was set by the International Telecommunication Union (ITU) Broadband commission. Furthermore, there is still an ongoing debate of policies and on how to expand broadband; while some nations such as the USA prefer the regulator to play a leading role, others such as South Korea provide public infrastructure support [4]. Nevertheless, International Mobile Telecommunications (IMT) 2020, with the marketing name 5G, is expected to play a great role in reaching this target. In addition, to meet the 2025 connectivity goal and other outlined targets, the commission recommended that each country needs a national broadband strategy and evaluation program to assess and/or

expand its internet access. Following this recommendation, many countries are conducting a yearly service quality assessment.

Currently, there are multiple options to deliver internet to users such as wired (copper or optical) and wireless (cellular/mobile broadband (for the rest of this report, cellular and mobile broadband are used interchangeably), or satellite). As can be seen from Table 1, provided by ITU, mobile broadband connections have become the main choice to access the internet and this trend is only going to continue thanks to the falling price of subscriptions. In addition to the lower cost, public support and ownership are also other important factors to the success of the mobile broadband connection [5].

**Table 1.** Percentage of world population with communication as of 2018.

| Indicator | 2018 Data | Penetration | Source |
|---|---|---|---|
| World Population | 7.6 billion | - | UN |
| Mobile Broadband Subscription | 5.3 billion | 69% | ITU |
| Unique Mobile Subscribers | 5.1 billion | 67% | GSMAi |
| Unique Mobile Internet Subscribers | 3.5 billion | 47% | GSMAi |
| Internet Users | 3.9 billion | 51 % | ITU |
| Active Social media Users | 3.5 billion | 45% | Datareportal (Hootsuite) |
| Fixed Broadband Subscription | 1.1 billion | 14% | ITU |

Mobile broadband services are provided via 2G, 3G, 4G, and 5G technologies. Until being overtaken by 4G in 2018, 2G was the most prominent mobile broadband technology. Currently, 4G connects over 45% of the total mobile broadband connections and it is expected to continue to grow up to 62% in 2023. Although 5G and its attractive applications such as IoT, autonomous cars, virtual/virtual reality, and smart home/cities are projected to overtake 4G by 2025, it might be a little delayed due to the delay of other technologies and the unprecedented pandemic in 2020. Furthermore, both nations and MNOs are still examining the economic impact or growth that will come together with faster broadband access [6]. Additionally, researchers fear that applications that require such speed are not here yet, which might render 5G a fancy privilege for a selected few [7].

Nevertheless, the deployment of 5G is still underway in countries such as Korea, the United States of America (US/USA), the United Kingdom (UK), Australia, Singapore, Germany, and France. Furthermore, over the next 5 to 10 years, mobile network operators (MNOs) all over the world are planning to spend about USD 1.5 trillion on infrastructure, which will facilitate the prominence of 5G [8].

Therefore, in this paper, we compare the mobile broadband quality assessment results obtained by regulators of countries such as the US, UK, Australia, Singapore, Germany, Japan, South Korea, and France. These nations are OECD members and reflect different parts of the world. Furthermore, the selection of these nations takes the highest and lowest population densities into consideration.

The report will allow:

- Consumers to comprehend and compare the quality of the communication service they are receiving as well as the government's/regulator's relentless efforts in ensuring a high level of service quality and guaranteeing investments to improve said quality for its citizens;
- Countries to continue as leading nations in communication service and meet the targets set out by the broadband commission;
- Mobile broadband companies to understand their status within and outside the country, which aids their R & D efforts to stay ahead of competitors such that the country meets its "5G+ Strategy";
- Regulators to find out new and suitable metrics and synchronize them.

The rest of this paper is organized as follows: In Section 2, the methodology of the study is discussed. The case studies from each nation are presented in Section 3. Section 4

is dedicated to examining the progress made towards 5G. Finally, the summary and conclusion of this paper are presented in the last section.

## 2. Methodology

There are a variety of methods and metrics to evaluate the quality of mobile broadband services. Hence, each country has used a subset of these methods and metrics for evaluations. In the future, these countries and the ITU are expected to harmonize mobile broadband service evaluations such that consumers realize that the service provided in their country is of the highest standard. One such effort is being made by the Body of European Regulators for Electronic Communication (BEREC) [9]. However, to uniformly summarize the quality assessment from the regulatory body of each country, this report follows the structure detailed below.

The summary of the regulatory report of each country is organized by:

- **Regulator**: The body responsible for mobile broadband service quality evaluation, the year the regulator started the assessment, the year regulators published the most recent report, and which year that report is about. For instance, Germany's last report was published in 2019 about the year 2018.
- **Evaluated service**: The mobile broadband service among 2G, 3G, or 4G is evaluated in the report. For instance, Singapore does not report on services such as 2G because it is no longer provided, whereas countries such as Japan discuss mobile broadband service in general rather than separately.
- **Operators**: The most prominent operators are mentioned in this part.
- **Method**: The method used to obtain the data and if statistical correction was used to account for errors is discussed here. Multiple means of obtaining data can be used.
- **Metrics**: As mentioned earlier, each country used different sets of metrics to evaluate the quality of their mobile broadband service. However, most of them provided some means of measuring coverage.
- **Results**: This topic is used to discuss the results obtained from each report and the policies outlined to improve the quality of the mobile broadband service.
- **Policies**: Policies outlined to improve the quality of mobile broadband service are presented.

## 3. Case Studies

The report from each nation's regulatory body contains results from multiple communication and infrastructure sectors. However, in this paper, we focus only on mobile broadband evaluation, and a summary from each report is presented. We looked at reports released in 2020 and some nations did not include current year data.

### 3.1. Broadband Deployment Report: USA

*Regulator* → The Federal Communication Commission (FCC) has published a yearly broadband deployment report on the quality of communication service for US consumers since the beginning of the 21st century. The 2020 report has a five-year evaluation from 2014 to 2018 included [10].

*Operators* → In the US there are four prominent operators providing mobile communication services: Verizon, AT&T, Sprint, and T-Mobile.

*Metrics* → To assess the deployment of advanced telecommunication for mobile services, which can reach a 5 Mbps download speed and 1 Mbps upload speed (5 Mbps/1 Mbps), via mobile broadband communication. However, the evaluation of 10 Mbps/ 3 Mbps access to mobile broadband services is also carried out. On the other hand, metrics such as latency is rejected due to the lack of reliable and sufficiently comprehensive data source for such analysis.

*Evaluated service* → In this report, the evaluation of only 4G LTE is included. For the 5 Mbps/1 Mbps evaluation, the country is categorized into rural, urban, and tribal lands, whereas 10 Mbps/3 Mbps do not include the third category.

*Result* →

- With the 5 Mbps/1 Mbps speed test, compared to 2014, rural and tribal lands have shown a 3% increase in geographical mobile service coverage in 2018, whereas there is an insignificant increase in urban coverage as almost all the area was covered in 2014.
- There is 97% of urban and 83% of rural areas coverage of 4G service with 10 Mbps/3 Mbps in 2018. However, this result is significant compared to 2014, which was just 70% and 80% coverage for rural and urban, respectively.

*Policies*→ The FCC acknowledges that there is still some work to be carried out to close the gap in coverage between rural and urban areas as well as the advertised and median speeds.

- The rule for an over-the-air reception device (OTARD) is expected to be updated such that 5G deployment can start.
- A Universal Service Fund (USF) to improve the rural area coverage is proposed.
- A new "Digital Opportunity Data Collection Order" data collection mechanism is adapted to determine the availability of broadband service.

Although most of the population living in rural and urban areas is covered by at least one of the MNOs, geographical coverage is still low.

*3.2. Evaluation Report: Korea*

*Regulator* → National Information Society Agency (NIA) is responsible for communication services performance evaluation in South Korea, which is referred to as Korea henceforth. The latest report is of 2018 about the evaluation of 2018. NIA has been composing this report since 1999 [11].

*Operators* → There are three prominent MNOs in Korea. LG U+, SK, KT.

*Method* → Several methods have been used by NIA. First, a professional evaluator moves around in a vehicle and by foot to measure performance through an app installed on evaluation devices. The locations of measurements were statistically sampled to represent the whole country and Galaxy Note 8 was the selected evaluation device in 2018. Second, an RF scanner tool is used to verify the coverage claim of MNOs. Third, a crowdsourcing-based evaluation from a statistically sampled population via an app is performed to understand customer satisfaction.

*Metrics* → Several metrics have been used to evaluate the mobile broadband service.

- Data/Internet
  – Connection success rate (SR) → The proportion of calls that attempted to access the measurement server and succeeded.
  – Transmission success rate (SR) → The proportion of calls that successfully connect to the measurement server and sent the data faster than the minimum speed.
  – Delay time → The time between signal transmission to the measurement server and receipt of an acknowledgement signal.
  – Packet loss rate →The proportion of the data that cannot be sent when sending and receiving the data between the device and measurement server of the service provider.
  – Transmission rate → Data sending and receiving speed between the device and measurement server of the service provider.
  – Web surfing time → The time required to display the entire web page on the device screen after the user inputs the website address.

- Voice
  – Successful connection → Success if connected within 20 s of pressing the call button.
  – Call drop → Dropped during a call (65 s).

- Poor sound quality → If the sound quality value (1–5 points) is below 2.2 or 1.9 more than two times in a row (2.2: difficult to understand, 1.9: impossible to understand).

- Coverage
  - Over-claim Rate → A proportion of the area where coverage opened by the communications service provider is overstated compared to the actual coverage information.

*Evaluated services* → While voices over 2G, 3G, and 4G services are evaluated, data and coverage are evaluated on the latter two services.

*Result* →

- The average download to upload speed stands at 150 Mbps by 44 Mbps, which showed 13 and 44 percent increases, respectively, compared to the year before. While the delay is below 40 ms on average, the loss rate is below 1%. All MNOs have considerably great services in terms of data via 4G/LTE services. This trend is similar for 3G services

- The crowd-sourced evaluation confirms the result obtained by the professional evaluator. The data collected via the app are categorized into five groups (A–E) based on maximum speed. The devices under group A have the same specification as that of the professional evaluator.

- The quality of voices is measured while using the same or a different company. The same company refers to the caller and receiver being on the same MNO as well as using the same device types whereas a different company refers to the caller and receiver being on the different MNO as well as using the different device types. Call success rate means the ability to make a successful connection under 20 s, no call drop for >65, and have a sound quality above 2.2. While the 4G-LTE service provides the highest quality, 2G provides the lowest, which is expected.

- While examining the MNOs coverage claim, the report indicated that the average overstatement is 12.9%. LG U+ has the highest overstatement at 34%, whereas SK and KT have 3% and 1% over claim rates, respectively.

The results indicate that all MNOs provide a great quality of service in Korea both in terms of data and voice. In addition, the results and claims are verified instead of being accepted as is.

*3.3. Connected Nations: UK*

*Regulator* → In the UK, OfCom, using MNOs, is the responsible organization for obtaining and compiling annual broadband service quality. The authority started compiling the report and service quality evaluation in 2004. A report was released in 2019, which is mainly focused on the years 2018 and 2019 [12].

*Operators* → According to the report, there are four well-known operators: EE, O2, Three, and Vodafone.

*Method* → The report is based on service for a premise, which is a household or a location where a person can receive mail. To avoid redundancy, only the head of the household is associated with a premise. Then, the premises are classified as urban and rural using a third-party data source. As of 2019, 30.8 million locations were identified as premises. The objective is then to determine if these premises are covered by one or all of the operators using the coverage data obtained from MNO [12] and prediction.

*Metrics* → In this report, the availability of voice, which means the ability to make a call for 90 s, and a data connection that guarantees 2 Mbps upload/download speed are measured. It is stated that 2 Mbps is fast enough to browse the internet and watch glitch-free mobile videos.

*Evaluated services* → The 4G service is mostly evaluated as consumers tend to predominantly use it these days. The report is organized by urban and rural areas. However, 2G and 3G mobile broadband services have also been evaluated.

*Result* →

- As of 2019, 4G is mainly used for carrying about 90% of all data traffic, while the rest is shared by 2G and 3G. On the other hand, 2G and 3G carried 6% and 73% of the voice traffic, respectively, while the remaining is served by 4G.
- Outside premise indicates locations where people usually live, work, or travel. It is assumed, by this regulator/OfCom, to be a good indicator of the availability of coverage.
  - The voice from all MNOs is at 99%, and from at least one MNO is 100%.
  - 4G data from all MNOs is about 97% available, whereas from at least one is about 100%.
- However, there is a significant difference between urban and rural areas.
  - The voice from all urban areas is at 100%, while that number is about 94% in the rural areas. Nevertheless, a voice from at least one MNO in rural areas remains at 100%.
  - With regard to 4G availability, in rural areas, at least one MNO it is as high as 99% and another is as high as 100%, whereas for urban areas, one is as low as 85% and one MNO is at least 99%.
  - Among MNOs, while EE has the best coverage in both urban and rural areas, Three has the least.
- Indoor/inside premise evaluation is presented in the report. Here, we present a summary of the indoor results as of 2019.
  - The voice from all is 93%, and from at least one is 100%. Here, O2 and Vodafone provide 99% of voice service, whereas EE and Three provide 96%
  - 4G from all is 80%, and 99% from at least one. In terms of 4G coverage, there is no significant difference from MNOs, but it is not specifically stated in the report.
- The indoor coverage results of the urban and rural gap in terms of both voice call and 4G coverage is very large.
  - Voice service from all urban areas is at 97%, whereas only 68% of indoor premises in rural areas benefit from this service.
  - This inconsistency becomes worse when considering 4G services. EE has better service in rural areas, whereas O2 and Vodafone provide a superior service in urban areas.
- Specific parts of the country that people often go to/use, such as roads, have also been checked for coverage. Motorways and A roads receive 81% of voice service coverage from all MNOs, whereas B roads only have 67% coverage. In terms of 4G service, the former receives 95% coverage, while the latter only receives 90%. This result is not broken down into rural and urban areas. Additionally, railroad coverage data are still under analysis.
- Thus far, the discussion focused on what percentage of the areas where people are present is receiving voice and 4G coverage. Actual physical land coverage is also presented in the report.
  - There is a 79% from all MNOs and 95% from at least one MNO of voice service in all of the UK's geographic areas. Vodafone and O2 have a 91% geographical coverage, whereas EE and Three are at only 86% for voice service.
  - As for 4G data service, the is 66% from all MOs and 91% from at least one operator. EE has the highest 4G geographical presence with 84%, whereas O2 has the lowest with 76%.

*Policies* → The UK government has negotiated with industries to overcome the gap in coverage between rural and urban areas through the "Shared Rural Network" (SRN). The proposal allows MNOs to share current and future infrastructures to improve the coverage in rural areas. In addition, an investment to create future-proofed 4G infrastructure and services will be made.

From the report, it is possible to observe that all operators had similar and good voice services in indoor premises, and O2 has come out as the winner for 4G data coverage. On the other hand, Vodafone and O2 have a better voice service for outdoor premises whereas the 4G data coverage from all MNOs is the same. Even though all operators have a high coverage for urban areas, the coverage for rural areas is low in almost all cases, which is why the government issued the SRN proposal.

*3.4. The State of the Internet in France*

*Regulator* → Autorité de Régulation des Communications Électroniques et des Postes (ARCEP) of France is the regulatory organization responsible for compiling a quality assessment of communication services; the assessment started in 1997 [13]. Starting in 2018, ARCEP provides an accompanying web for its report and evaluation (QoS Web App accessed on 26 Janaury 2022) and the latest date of evaluation available on the web as of 2019.

*Operators* → There are four major MNOs included in the report: Bouygues, Free, Orange, and SFR.

*Method* → The authority used to depend on on-site measurements by selecting locations using statistical methods such that it reflects the status of communication in the country. However, in recent years, ARCEP introduced a new assessment model that involves data provided from MNOs and crowdsourcing. *Metrics* → There are about 7 metrics in this report.

- Availability of voice → making a phone call for >2 min;
- Availability of Short Message Service (SMS) → receiving a message in <10 s;
- 4G data.
    - Web success rate → loading the 30 most visited websites in <10 s;
    - Download speed → receiving 10 Mbit data in <1 min;
    - Upload speed → sending 2 Mbit data in <1 min;
    - Streaming success rate→ watching a high definition (HD) video for >2 min with perfect quality.

*Evaluated services* → Although there are few 3G measurements, specifically for geographical coverage, almost all the assessment focuses on 4G services. The authority categorizes France into rural, suburban, urban, and tourist destination areas. The data are further refined by indoor, outdoor, and road coverages. For the sake of simplicity, tourist areas are skipped, since these areas are part of the other three, and road data are left out because they are incomplete.

*Result* →

- In all the metrics, the geography and population coverage of France, for both voice and 4G data, from at least one operator is >99%.
- The voice and SMS success rate of all four MNOs in France in all areas are 91% and 94%, respectively. These numbers, however, vary in rural and urban areas. For instance, the success rate of making a call that is >2 min with perfect quality in all indoor areas is 90%, whereas it is at 84% in rural areas and 94%.
- For 4G mobile broadband services, the average download and upload speeds in indoors are 39 Mbps and 7 Mbps, respectively. The average download and upload speeds indoors are 50 Mbps and 13 Mbps, respectively. As expected, there is a difference between urban, suburban, and rural areas. Furthermore, Orange provides the best 4G service; Free is the least effective.

*Policies* → The authority had plans to incorporate third-party quality assessments in its future reports, given that the data provided from these organizations follow a code-of-conduct guideline provided by ARCEP. A test for determining if an MNO satisfies the minimum requirement of each metric was considered reliable if it is >95%. However, ARCEP updated this number to >98% to force MNOs to provide a higher-quality service to consumers.

Although the 2020, The state of the internet in France, report does not offer a detailed quality evaluation, ARCEP, which is the organization responsible for communication service regulation and evaluation, provides a web app to assess the quality of service in the country (monreseaumobile.fr). Although most of the population receives an adequate mobile broadband service, with the best service from Orange and worst service from Free, ARCEP is demanding MNOs to step up their service by raising the reliability threshold.

*3.5. Annual Survey on InfoComm Usage: Singapore*

*Regulator* → Since 1990, the Infocomm Media Development Authority (IMDA) of Singapore has been conducting quality assessments and studies. Although the results were published in the survey [14], technical results are in the accompanying website (QoS Reports accessed on 26 January 2022). Although the report was released in 2019, the evaluation available on the web is of 2020.

*Operators* → There are three prominent mobile broadband service providers that are included in the report: M1, Singtel, and StarHub

*Method* → Around 6000 Singaporeans have been surveyed in 2019. The survey is conducted to represent only citizens and permanent residents as well as the number of individuals living in a single household. Data were collected via face-to-face interviews rather than online. In addition, on-field measurements of signals by professionals are also used. Although IMDA mentioned another specification document to be used together with the survey it was not stated which document.

*Metrics* → The metrics used in the IMDA report are organized as follows:

- Voice

  - Outdoor/Tunnel coverage → SNR > −100 dBm;
  - Call success rate (SR) → connecting to calling or getting a busy tone;
  - Call drop rate (DR) → unintended disconnection during >100 s call;
  - Indoor coverage → > 85% coverage per building.

- Data

  - Outdoor/Tunnel coverage → SNR> −109 dBm;
  - Indoor coverage → < 85% coverage per building.

*Evaluated services* → In Singapore, 3G is used for voice and SMS, whereas 4G is for data services as this is a small nation the report does not refine the data based on rural and urban areas.

*Result* →

- According to the IMDA's 3G Public Cellular Mobile Telephone Service (PCMTS) QoS framework, MNOs' outdoor coverage should be >99%. That is, the SNR received from an MNO by stations in any outdoor area in Singapore should be >−100 dBm. As of 2019, all three MNOs provide >99% voice service coverage. For tunnels, only a pass or fail is given based on the services provided and all three MNOs passed in all tunnels.
- Similarly, a 99% call success rate is expected using the PCMTS framework, which all three MNOs fulfilled.
- The call drop rate should be <1% under the PCMTS framework, which none of the MNOs are above.
- Sixty buildings were selected to test the indoor services. Even though the data from the fourth quarter of 2019 are not completed yet, all three MNOs have an average score of 59/60. This means there is 99% indoor voice service coverage.
- As for 4G service, all three prominent MNOs and one new operator (TPG) are evaluated. The 4G service QoS framework was lowered for TPG since it is new. However, all four MNOs provide >99% outdoor service, have a pass mark in all tunnels, and have an average score of 58/60 in indoor coverage.

*Policies* → IMDA sets a QoS standard and updates it regularly. An update is issued when there is industrial, technological, or consumer demand change. MNOs are expected to fulfill the QoS requirement or receive a USD 50,000 fine.

The results from the IMDA report are few as Singapore is a small nation. In addition, all MNOs including the new one provide excellent mobile broadband service as there are few areas to cover.

### 3.6. Communication Report: Australia

*Regulator* → Australian Communications and Media Authority (ACMA) of Australia also used survey-based quality evaluation. The authority started gathering and analyzing the report in 2005. The report was released in 2020 with 2018–2019 evaluation results [15].

*Operators* → There are three major MNOs in the country: Telstra, Vodafone, and Optus.

*Method* → Each year since 2005, ACMA surveys a group of people using statistical sampling to represent the population. As of 2019, the Australian adult population aged >18 stands at 19.4 million, and accordingly, 2067 people have been surveyed for this fiscal year. The data from the survey are then weighted to reflect the population. However, the annual report is based on the number of users instead of coverage.

*Metrics* → Although there are no specific metrics that are useful for technical evaluation, ACMA together with AMTA requested all MNOs to provide a coverage map. This coverage map will represent the population covered instead of the geographic area covered. However, here are some of the metrics used to evaluate the quality of mobile broadband services:

- Customer satisfaction → rating out of 5;
- Pre- vs. postpaid service;
- Market share.

*Evaluated services* → 3G, and 4G, mobile broadband services coverages are expected to be available at each MNOs website. The data are then expected to be refined by area.

*Result* → Even though it is not technical, here are some data extracted from the report.

- Coverage of population from at least one MNO for 3G and 4G services reached 99.4%. In addition, there are now >400 5G-capable base stations in Australia.
- Among the total population, 96% have mobile phones and 83% have smartphones. There are around 35.9 million mobile subscribers from all MNOs, where 54%, 29%, and 15% of the total subscribers belong to Telstra, Optus, and Vodafone, respectively.
- Among the subscribers, 69% are postpaid while the rest are prepaid. This number is only slightly different from previous years.
- There are arrays of quality of services evaluated through customer satisfaction. Although coverage satisfaction has improved, it is still under 4. The same is true for data speed and voice quality.

*Policies* → A revised consumers protection code is issued to protect consumers from bad selling practices, credit assessment, financial crises due to communication services. In addition, a revised mobile base station deployment policy is also issued to strengthen communication services in rural and under-served areas.

The report by ACMA mainly focused on the market share of MNOs and revenues from communication services, which made it difficult to present more results. Although the authority demanded MNOs to provide the coverage map, the data available in each MNO are not sufficient enough to display.

### 3.7. White Paper on Information and Communications: Japan

*Regulator* → As to the state of communication in Japan, the Ministry of Internal Affairs and Communications (MIC) provides the evaluation report. The MIC of Japan's latest report was released in 2019, with most of the results covering up to 2018 [16]. The report dates back to 1993 in some different forms.

*Operators* → The three major mobile broadband service providers in Japan are NTTDO-COMO, KDDI, and SoftBank.

*Method* → There is no clear statement in the report. The results indicate survey and MNO data have been used.

*Metrics* → Most of the metrics presented in the report are not technical.

- Number of subscribers;
- Average revenue per user (ARPU);
- Fixed vs. mobile communication services;
- Data vs. voice services;
- Market share.

*Evaluated services*→ 3.9G or broadband wireless access (BWA) and 4G service.

*Result* →

- The number of BWA users is estimated to be >136 million and 4G subscribers are more than 66 million.
- While the percentage of fixed-line communication sales has been shrinking over the years, those of mobile broadband have been growing.
- Although yearly ARPU is increasing, KDDI and SoftBank registered the highest and lowest ARPU, respectively.
- Voice service shrunk over the years while the usage of data services increased significantly over the years.
- As of 2019, NTT DOCOMO has the highest mobile subscribers market share with 39% total market. KDDI and SoftBank have 27% and 23% shares of the market, while the rest is shared among the subsidies of the three companies.

*Policies* → A new comprehensive verification of competition rule is proposed by the organization to ensure the growth of telecommunication. The rule requires MNOs to have (1) local and global vision, (2) have enhanced communication maintenance, (3) maintain net neutrality, and (4) address future IT platform services.

The report provided by MIC is exhaustive from a market metrics point of view. However, when it comes to technical mobile broadband service assessment and evaluation, it does not provide comprehensive results. Although a thorough investigation of other organizations that also provide a quality evaluation of mobile broadband services, such as Statistics Bureau (Japan e-stat accessed on 26 January 2022) and Ministry of Economy, Trade and Industry (Japan meti accessed on 26 January 2022), a useful technical evaluation result could not be obtained.

### 3.8. Annual Report, 20 Years of Responsibility for Networks: Germany

*Regulator* → The Bundesnetzagentur organization, starting in the early 2000s, is responsible for measuring and assessing the quality of mobile broadband service in Germany. Bundesnetzagentur is also responsible for assisting energy, transportation, and postal service qualities. The latest report was released in 2019 [17]. However, the evaluation results are up to the year 2018.

*Operators* → Although Deutsche Telekom AG (DTAG) is the oldest MNO in Germany, there are also two more major MNOs: Vodafone and Telefonica Germany.

*Method* → Data gathered from MNOs and surveys by Bundesnetzagentur are used to compile the data in the report. Although there is no section in the report that specifically discusses the method, the results and the explanation implied where the data for the report are collected from.

*Metrics* → There are simple technical metrics and other non-technical metrics used in this report.

- Coverage → from at least one MNO and each MNO;
- Speed → download;
- Applications → Messages sent, call minutes, and data volumes;
- Revenue per sim card;
- Number of subscribers.

*Evaluated services* → Although applications provided via 2G, 3G, and 4G are evaluated, the specific services used for each application are not mentioned.

*Result* →

- In 2018, the 4G data service is available for 97.5% and 95.5% of the population with 2 Mbps and 6 Mbps download speed from at least one MNO. When considering individual MNOs, DTAG, Vodafone and Telefonika Germany provide 98%, 93%, and 88% coverage, respectively, which means the service from all MNOs is 95%.
- More than 1993 m GB of data were sent through mobile broadband communication using 6.78 million UMTS- and LTE-enabled devices.
- On the other hand, while the calls made via mobile broadband services are on the rise and surpassing fixed network calls, the number of SMS is on the decline. In the year 2018, around 119 billion minutes and >8.9 billion messages were sent.
- While the revenue from data service is on the rise, the revenue generated from voice and SMS remains constant when compared to the past data.

*Policies* → When allocating a new spectrum, a new coverage regulation that obliges MNOs to provide access to data rates of >10 Mbps for each household was issued. To close the coverage gap, MNOs are required to install new base stations at roads and railways as well as not-spots.

## 4. 5G Communication Status

When discussing 5G, which is the marketing name for IMT-2020, demystifying the misconception that technology is an upgrade to its predecessor 4G/IMT-Advanced, which depicts 5G as an enhanced mobile broadband communication technology [18,19]. Additionally, 5G is an ecosystem that encompasses several applications and services that are expected to empower authorities as well as companies to be able to transform their cities and industries, respectively. In order to accommodate new and future applications, 5G's capacity has to be improved in many dimensions. In Reference [20], the authors go as far as suggesting that 5G needs to be supplemented with other communication technologies such as Wi-Fi. Figure 1 shows the enhancements expected of 5G in comparison to its predecessor, 4G.

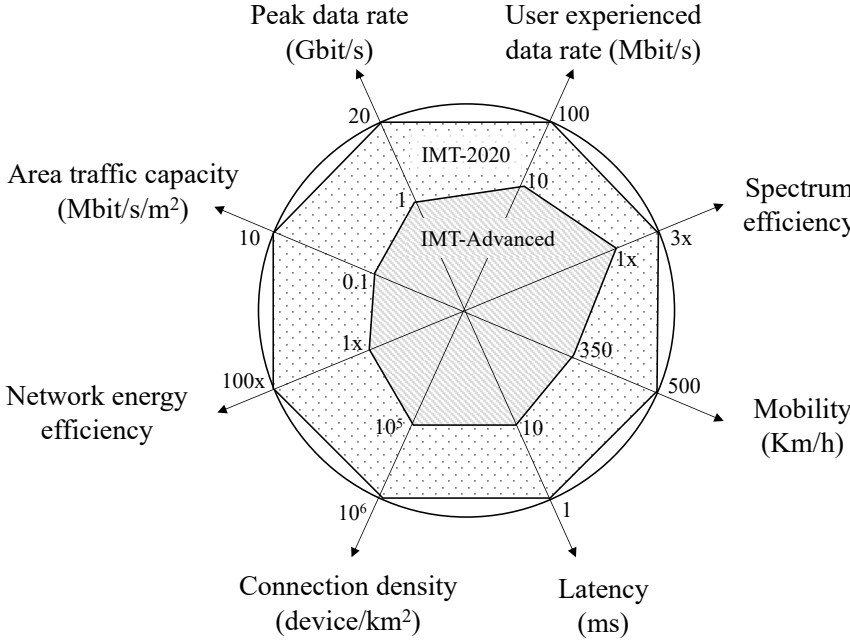

**Figure 1.** IMT-2020 (5G) vs. IMT-Advanced (4G) [2].

In addition to technical enhancements, there are about 11 general requirements to be addressed before the rollout of the technology [2,18]; 5G is expected to: [18,21]

- Support service diversity in terms of QoS, traffic patterns, mobility and/or data types;
- Some of these services are incompatible and require 5G adaptable via programmability;
- Needs to serve new radio technologies and work with the old CN;
- Allow distributed network architecture unlike its centralized predecessors;
- Have separation of control and user plane functions;
- Process and store data to reduce congestion and response time;
- Possess intelligence such that automation and interoperability can be implemented;
- Serve a user demand with optimal capacity;
- Guarantee reliability with sufficient performance;
- Provide security and privacy to devices as well as data;
- Be energy efficient.

With such a meticulous list of requirements together with enhancements in all directions, 5G can maintain several applications and services. However, evaluating the 5G services with uniform metrics does not reflect whether the quality of the service is acceptable or not. Hence, metrics suitable for each service must be devised. Another issue that needs to be addressed is the effort of countries to promote and facilitate the move towards 5G. These and related issues are discussed in the following subsections.

### 4.1. Applications and Services

Although there are numerous applications that are to come out of 5G, there are three major categories all these applications and services can be classified into, Figure 2. The classification is based on the requirements of each application such as number and type/resource of devices, mobility, latency, and speed [18].

- **Enhanced Mobile broadband (eMBB)** → Traditional mobile broadband applications such as Voice, SMS, and data services fall into this category. While the major new application that can fall in this category include Augmented Reality (AR), Virtual Reality (VR), and Gaming [18,22].
- **Massive machine-type communication (mMTC)** → Mostly, IoT together with smart home, city, and agriculture are part of this class [18,23].
- **Ultra-reliable and low-latency communication (URLLC)** → Extremely delay-sensitive applications such as e-health, autonomous vehicles, and industrial automation are associated with this class [18].

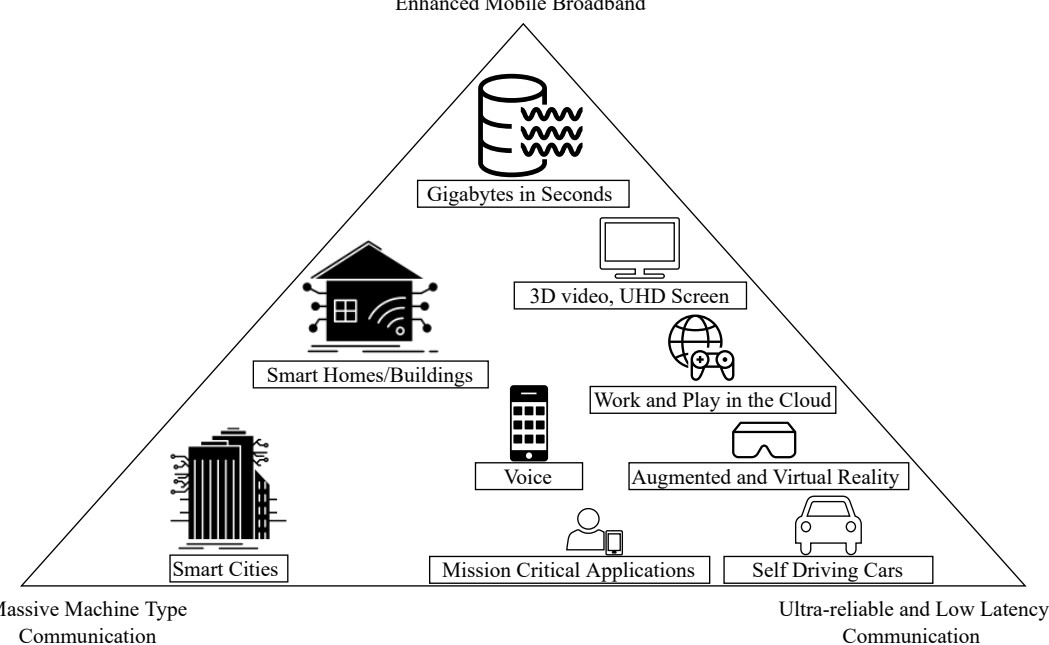

**Figure 2.** Applications and services in IMT-2020 (5G) [2].

Approaching all applications with similar network infrastructure is wasteful and costly. For instance, mMTC applications have small data (in Kbps), which means allocating a resource that is Gbps speed is needless. Hence, 5G applies a new network architecture, which is network slicing [24]. Right now, there are four Slicing/Service Types (SSTs) under the 3GPP document for the 5G specification [25]. The SST values are 1, 2, 3 and 4, for eMBB, URLLC, mMTC, and V2X applications, respectively.

*4.2. Evaluating 5G Performance*

Understanding the characteristics of 5G as well as its applications and services before attempting to evaluate its performance is critical. Hence, in this section, we first investigate what is different in 5G and then point out how to evaluate its performance.

Coverage, be it geographical vs. population, indoor vs. outdoor, urban vs. rural, or any other way, will remain one of the key metrics in evaluation 5G [26]. There are inherent problems that are associated with evaluating coverage even when one is dealing with only mobile broadband services operating in narrow bands. Meanwhile, 5G is going to make use of mmWave and other unconventional bands, which will make evaluating coverage a more complicated task. To emphasize that a summary of coverage by Signals Research Group (SGR) is presented here.

The first task the group undertook is to compare the performance of 5G with its predecessor 4G in different situations and countries. In Korea, the testers drive around COEX mall at the night and walked during the afternoon as well as evening [26,27]; 4G and 5G phones, under SK telecom, were used to test the performance. In addition, adjustments, to reflect the resource blocks (RBs) assigned to the phones, are made. The raw and the normalized results both confirmed that there is a faster speed when using 5G phones, 2.6× times faster to be specific.

Similar measurements by SGR were conducted in London, UK, Bern, Switzerland, and Minneapolis, US, via EE, Swisscom, and Verizon operators, respectively. All tests confirmed that 5G networks provide a faster speed than that of 4G.

However, this is expected and obvious and there are other major surprises in the study. The first one comes while testing mmWave [26]. Although the researchers were facing in the opposite direction of the 5G transmitter and there was no direct transmission to the phone, the 5G phone received a signal as well as relevant throughput, which was unexpected. It was later discovered that such a phenomenon occurred due to the reflection of mmWave signal on nearby glasses, which led the group to conclude the unique behavior of mmWave requires further investigation.

Another unexpected result was observed during the analysis of the signal quality of 5G mmWave while placing the phone in landscape or portrait mode. Both the beam reference-signal-received-power (BRSRP) and the signal-to-noise ratio (SNR) exhibited a better performance in portrait mode. On the theme of phone behavior towards mmWave, another test where the phone was placed below two hands (death-grip) showed that there is performance degradation both in terms of BRSRP and SNR. It was clear that even in that scenario 5G with mmWave shows a better result than 4G.

These and other results from the study suggested that the signal strength, which in turn may affect the coverage quality evaluation, from 5G mmWave is stronger and unique than anticipated. Therefore, better coverage analysis methodologies shall be used in the future. One such approach is to enlist robust pieces of equipment such as unmanned aerial vehicles (UAVs), as proposed here [28].

*4.3. Metrics for 5G*

As indicated above, 5G consumers as well as applications are diverse and should be treated differently by the eNBs. Not only this, but service quality and performance evaluations for each class should also be carried out separately. Furthermore, applications might have behaviors of multiple classes, in such case, the evaluation can include metrics from both. For instance, AR requires low latency and faster speed, which means metrics

from URLLC and eMBB can be applied to evaluate the service provided for this application. In addition, different performance requirements are also set out for each class. Therefore, metrics suitable for each class should be discussed separately.

- **eMMB metrics** → The traditional metrics that are discussed above, which were used to evaluate the performance of mobile broadband communication, can be harmonized, and new methodologies such as the usage of UAVs can be incorporated to evaluate eMBB
- **mMTC metrics** → Research on metrics used to assess the quality of IoT is presented in later sections, which mostly can also be applied for mMTC.
- **URLLC metrics** → As discussed, the unique behavior of applications requires new and innovative metrics to evaluate applications and services under this class.

*4.4. Status by Country*

Since 5G has recently been deployed and most of the reports compiled previous years data, there is very little information as well as a quality assessment conducted by regulators. Rather, there is some information on the status and progress towards 5G, which is discussed here.

- *USA* → Three out of four major operators, AT&T, Sprint, and Verizon, had plans to deploy 5G networks in multiple cities starting in late 2018.
    – States have announced a public–private partnership to improve the deployment of 5G and other broadband communication infrastructures.
    – Recent press releases by FCC indicated the availability of a new spectrum for 5G, which means more deployments to follow [29].
- *UK* → Ofcom claimed the UK was the leader of 5G in Europe as all four prominent MNOs started providing the service in more than 40 cities and towns [30]. Ofcom is also exploring new schemes to promote 5G.
    – A new approach for businesses and organizations to access operator or self-deploy 5G network to endure security and privacy has been provided [31].
    – In addition, spectrum licensing and sharing guidelines to support this approach are also outlined [30].
- *France* → There is very little in the report from ARCEP, which could be because France is planning to launch 5G in the year 2020.
    – One unique requirement for 5G spectrum holders is that MNOs should deploy IPv6 compatible to ensure future-proof infrastructure.
- *Singapore* → There are several efforts currently underway IMDA to begin the deployment of 5G by 2020 [32].
    – 4G TDD networks are ordered to have stringent synchronization to minimize interference such that there is minimal use of inter-operator guard bands.
    – Spectrum sharing as most of the spectrum occupied by 4G continues to be so for the coming few years.
- *Australia* → Two major MNOs, Telstra and Optus, have rolled out 5G, starting in 2019. Currently, there are more than 400 5G enabled sites in Australia. Compared to the rest of the nations, there are more developments in the 5G arena in Australia [15].
    – A communication standard to support connected and autonomous vehicles is introduced.
    – A five-year, 2018–2022, spectrum management plan that promotes 5G was released.
    – All major operators issued plans to start rolling out 5G by the year 2020.
- *Japan* → MIC considers 5G as an ICT infrastructure that should not only cover residential areas but also be available nationwide. 5G was expected to be available in Sep. 2019 [16].

- – Contests to promote and generate ideas on the usage of 5G were held in 2019.
- *Germany* → As Bundesnetzagentur expects the rollout of 5G in 2020 and 2021, only spectrum regulations are in place so far, [17].
  - – Following old regulations, MNOs are expected to provide >97% coverage of rural and urban areas as well as transport routes.
- *Korea* → All three major MNOs in Korea have rolled out 5G in 2019. Although there is no resounding public reception as its predecessor 4G.
  - – The regulatory body is also working on how to evaluate the different services that come with 5G.

One common theme is that MNOs from the countries included in this report have started or will start rolling out 5G beginning the year 2020. However, let us also not forget that this process will be hampered by the COVID-19 pandemic, which means it will take more time to realize the full impact of 5G than expected.

## 5. Discussion and Conclusions

In this section, a summary of the case studies, discussion, and conclusion is presented. Most of the regulators started performance evaluations in the late 1990s and early 2000s, except for Japan, which started earlier than most, Table 2. However, the Japanese evaluation is mostly non-technical. While Australia measures the performance of mobile broadband as a whole, the US only provided 4G measurements. On the other hand, countries such as Korea and the UK conducted separate and combined (Comb.) performance evaluations. In most cases, while 2G and 3G are for voice and SMS, 4G is used to deliver data services. None of these countries in this study conducted a 5G performance evaluation. Although the reports were supposed to drive the improvement of mobile broadband services, the starting year of the compilation of reports has no correlations with the current rank of the nations. On the other hand, the fact that most of these nations are working to provide real-time service assessment via web apps is encouraging. However, access to the web apps is not straightforward, and there is a lack of awareness by the public, which regulators need to address.

Five of the eight nations have three MNOs, whereas the rest have four MNOs (Table 3). Only one, Vodafone MNO, provides services in three different countries. Although Telstra is present in Singapore, its portion of the market and coverage is insignificant. The rest of the MNOs provide service only in a single country. This indicates, the geographical area and population size of the country do not have an impact on the number of MNOs per country. In addition, countries still prefer local MNOs, this is due to the inherent privatization of what used to be national MNOs [33].

**Table 2.** History of mobile broadband performance evaluations.

|  | Regulator | Evaluated Service | Starting Year | Last Year | Recent Report |
|---|---|---|---|---|---|
| Australia | ACMA | Comb. | 2005 | 2019 | 2020 |
| France | Arcep | 3G, 4G, Comb. | 1997 | 2019 | 2018 |
| Germany | Bundesnetzagentur | Comb., 4G | 2000 | 2018 | 2020 |
| Korea | NIA | 2G, 3G, 4G, Comb. | 1999 | 2018 | 2018 |
| Japan | MIC | 3G, 4G, Comb. | 1993 | 2018 | 2019 |
| Singapore | IMDA | 3G, 4G | 1990 | 2020 | 2019 |
| UK | OffCom | 2G, 3G, 4G, Comb. | 2004 | 2019 | 2019 |
| US | FCC | 4G | 2000 | 2018 | 2020 |

**Table 3.** Mobile network operators per country.

|  | Total | Major MNOs |
| --- | --- | --- |
| Australia | 3 | Telstra, Vodafone, and Optus |
| France | 4 | Bouygues, Free, Orange, and SFR |
| Germany | 3 | Deutsche Telekom AG (DTAG), Vodafone, and Telefonica Germany |
| Korea | 3 | SK, KT, LG U+ |
| Japan | 3 | NTT DOCOMO, KDDI, and SoftBank |
| Singapore | 3 | Singtel, M1, and  StarHub |
| UK | 4 | EE, O2, Three, and Vodafone |
| US | 4 | Verizon, T-Mobile, AT&T, and Sprint |

There are several ways to obtain data for performance evaluation. Most nations used coverage and services data provided by MNOs or surveyed the public (Table 4). However, Korean and Singaporean regulators conducted a field survey by professional evaluators, which measured metrics such as the received signal strength and call success rate. After obtaining data, some nations applied statistical correction to account for errors. The UK is the only nation that included predictions based on past data in its evaluation. Additionally, the data from third-party performance testers are also used in some cases. We believe, a combination of these techniques is necessary to provide an accurate assessment. Furthermore, we recommend obtaining data from public surveys, MNOs servers, and professional evaluators, which can be fed into a machine-learning algorithm to obtain better insights [34].

There is no standard broadband coverage evaluation, rather each nation provided its own version (Table 5). Some nations consider coverage as a percentage of the population receiving mobile broadband services; others define coverage where the signal is received (blanket coverage). All nations provide what percentage of the consumer receives mobile broadband coverage by at least one MNO. While the rest of the nations provided coverage from all MNOs, Australia and Japan did not disclose this figure, which is also true for coverage reports from each MNO. A very detailed coverage report such as indoor, outdoor, rural, or urban areas is only available on the UK and France reports. While blanket coverage indirectly guarantees population coverage, it is not feasible for countries with a wide geographical area. Instead, nations should improve the percentage of population covered by all MNOs, as it is very low compared to the percentage of the population by at least one MNO. Furthermore, indoor, outdoor, rural, and urban coverage evaluations with all MNOs need to be investigated for better insights.

**Table 4.** Data-obtaining strategies.

|  | On Field | MNO | Survey | 3rd Party | Prediction | Stat. |
| --- | --- | --- | --- | --- | --- | --- |
| Australia |  |  | X |  |  | X |
| France |  | X |  | X |  | X |
| Germany |  | X | X |  |  |  |
| Korea | X | X | X | X |  | X |
| Japan |  | X | X |  |  |  |
| Singapore | X |  | X |  |  |  |
| UK |  | X |  |  | X |  |
| US |  |  | X | X |  | X |

**Table 5.** Broadband coverage report.

|  | One | All | Indoor | Outdoor | Urban | Rural | V/S | Data | Pop. | Geo. | Roads | Each MNO |
|---|---|---|---|---|---|---|---|---|---|---|---|---|
| Australia | X | | | | | | | | | | | |
| France | X | X | X | X | X | X | X | X | X | X | X | X |
| Germany | X | X | | | | | | | | | | X |
| Korea | X | X | | | | | | | X | X | | X |
| Japan | X | | | | | | | | | | | |
| Singapore | X | X | X | X | | | X | X | | X | X | X |
| UK | X | X | X | X | X | X | X | X | X | X | X | X |
| US | X | X | | | X | X | | X | X | X | | X |

A summary of all other metrics used by the eight countries is presented in Table 6. These metrics are subjective such as satisfaction of consumers as well as objectives such as download and upload speed. In countries such as Germany and Japan, voice and data usage per annum is considered a factor. On the other hand, Singapore, Korea, the UK, and France focus on the application success rate. The US is the only nation that used the churn metric. Among these metrics, instead of economical factors such as revenue, regulators need to focus on technical aspects such as speed, success rate, and churn.

**Table 6.** Other types of metrics to evaluate mobile broadband.

| Metrics | | Australia | France | Germany | Korea | Japan | Singapore | UK | US |
|---|---|---|---|---|---|---|---|---|---|
| Churn | | | | | | | | | X |
| Subscribers | | X | X | X | X | X | X | X | X |
| Speed | Down. | | X | | X | | | X | X |
| | Up. | | X | | X | | | X | X |
| Video SR | | | X | | | | | X | |
| Voice SR | | | X | | X | | X | X | |
| Web SR | | | X | | | | | | |
| Voice DR | | | | | | | X | | |
| Voice/Year | | | | X | | X | | | |
| Data/year | | | | X | | X | | | |
| Satisfaction | | X | | | | | | | |
| Post. vs. Pre. | | X | | | | | | | |
| Revenue | | X | | X | | X | | | |
| Others | | X | X | X | X | X | X | X | X |

Finally, here are some of the plans, to improve mobile broadband services, forwarded from surveyed countries:

- Universal fund to improve coverage in rural areas;
- Push MNOs to share infrastructure in rural areas to improve coverage;
- Include and validate third party evaluation to improve quality assessment;
- Create penalties for MNOs that does not deliver on promised quality;
- Create consumer protection code/document;
- Demand MNOs to build future proof infrastructure;
- Installing more base stations in and around transportation facilities.

During this investigation, it was observed that each country has its own format for evaluating the performance of mobile broadband services. This could be due to the various metrics available to evaluate the service and the choice made by the regulators. However, some countries such as Germany and Japan were looking to evaluate mobile broadband services in terms of economic metrics, which conforms to the evaluation methods of other infrastructures. However, this could be avoided in future evaluations, since organizations such as BEREC of EU are working in unified metrics. The characteristics of 5G, discussed above, could further complicate the process of determining unified metrics.

Furthermore, we believe that verifying data obtained by one means through other means can also bear useful insights. For instance, in Korea, evaluations by professionals are validated via crowdsourcing results, and the coverage from MNOs is verified via professional evaluators from the regulator. On the other hand, France, and UK regulators provide a piece of exhaustive graphical information through a website. However, such information for the untrained eye is very confusing. Since one of the main goals of such a performance evaluation is to provide information to consumers such that they can make an informed decision when purchasing their next service, regulators need to provide straightforward information suitable for consumers.

**Author Contributions:** Conceptualization, methodology, validation, and writing—original draft preparation, Y.Z.J., W.-y.J. and B.K.; supervision, project administration, funding acquisition, B.K. and W.-y.J.; software, resources, visualization, Y.Z.J. and M.A.; writing—review and editing, validation, M.A. and R.P. All authors have read and agreed to the published version of the manuscript.

**Funding:** This research is supported by the Bisa Research Grant of Keimyung University in 2020 (No. 20200619).

**Data Availability Statement:** The data used in the experimental evaluation of this study are available within this article.

**Conflicts of Interest:** The authors declare no conflict of interest regarding the publication of this manuscript

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
