# Peer review of "Mobile Broadband Performance Evaluation: Analysis of National Reports"

_electronics, doi:10.3390/electronics11030485_

Round 1

Reviewer 1 Report

This professional paper aims at providing an analysis of national reports in terms of mobile broadband performance evaluation. The authors have addressed an important need to compare “the mobile broadband quality assessments obtained by regulators of countries such as US, UK, Australia, Singapore, Germany, and France”. Although the authors have covered rich information based on the reports from different countries, there is a critical lack of an analysis framework to make this analysis systematic and useful. That is why I would recommend the authors to improve the manuscript significantly and make their work valuable to other scholars in the field.

The manuscript is more like a descriptive work rather than an analytic and reflective work. Other than giving a description of reports from different countries, what would be the key motivation of doing this review? What would be the theoretical framework that drives the authors to do this review? After doing this review, what are the findings (including implications and limitations)? What are the key areas of research that have not been addressed in the literature? There are some preliminary ideas in the manuscript but the discussion section, for example, looks more like a summary of the information described in the previous sections, rather than an analysis and a reflection on the reports reviewed.

There is a lack of a set of selection criteria in the review work. What are the criteria and the process the authors used to decide which reports should or should not be included in the review, so as to make the review reliable and valuable?

Here are some other comments that could help improve the manuscript. The methodology used in this research should be described instead of describing the structure of regulatory reports of each country.  The references should be updated with the most recently published reports (e.g., authors used FCC report published in 2020, although there is FCC report from 2021: https://www.documentcloud.org/documents/21064816-fcc-broadband-deployment-report-jan-2021), as well as extended with the relevant scientific references. There are too many abbreviations (both in the abstract and text) that should be described at first mentioning, etc.

Author Response

Subject: Submission of review of the manuscript (electronics-1514655).

We are enclosing herewith resubmission of the manuscript entitled Mobile Broadband Performance Evaluation: Analysis of National Reports” submitted as a technical paper in MDPI Electronics Special Issue “Wireless Network Protocols and Performance Evaluation, Volume II”.

In accordance with the editor’s and reviewers' suggestions, we would like to notify you that this paper is a resubmission of a previous paper with ID number (electronics-1514655). We are very grateful for the comments as well as the time that the reviewers and editors have put into improving this work.

The reviews are incorporated in our modified version and hopefully, now our manuscript is more detailed and descriptive as compared to the previous version. Modified/Added contents are BLUE in color in the new version of the submission to illustrate the.  Attached please find our answer to the reviewer’s comment.

Best regards,

Yalew Zelalem Jembre (Ph.D.) 

Assistant Professor 

Department of Electronic Engineering 

Keimyung University, 

Reviewer 2 Report

Dear Authors,
Reading your work, I am convinced that you have done a conscientious and good job. It was particularly interesting to read the data from many different countries. 
Unfortunately, however, you have made some mistakes in the format and comparisons, which I would ask you to correct.
While reading the technical and scientific content is palpable, I do not find deep comparisons.
Unfortunately, the work contains a many formal errors, which could easily be corrected.

The abbreviations are not always resolved in the first place. Please check the document again. Punctuation at the end of sentences is missing in many places.

I collected several minor and major errors:

P1, L16, "The internet has come a long road since its inception back in 1969"- Please add citation.
P1, L20, "74 thousand Gigabytes"  - Please add citation.
P2, L34-35 "wired (Ethernet, coax, or fiber)" - Please consider to use "copper or optical".
P2, L54 "1.5 trillion USD in infrastructure"   - Please add citation.
P2, L58-59, Please rephase the sentence.
p2, L59, Please correct "5g" to 5G.
P2, L71, "The rest of this paper is organized as follows:" I can not see any description here. Please correct.
P2, L75-77, "In the future, these countries and the ITU need to harmonize mobile broadband service evaluation such that consumers realize the service provided in their country is of the highest standard or demand otherwise."
Please add pros and cons. If it is a hypothesis, please proof it.
P3, L100, please add "."
On Page 3, the first word of the list uses different format. As L90 and L111"Operators:" Please use the same format and fonts in all sections.

P7, L306/307. Please consider to use MB or Mb as, "MByte or Mbit".
P11, L478 "Metrics" Located in the middle of the row.
P13, L576, "study. The first one comes while testing mmWave", Please add citation.

In the Discussion and Conclusion section, the authors highlight the differences between mobile operators in many countries around the world. The Conclusion is a significant added value of the work and could be a major driver and driver for a new and scientifically reflective article. 

Please consider to increase the number of the references. 27 reference for a 18 page survey article is not suitable.
Authors shall use more specific references. As reference number 6-7-8 can not be easily reached. Please add more specific name of the document, or online address.

Author Response

(The authors gave the same response as above.)

Round 2

Reviewer 2 Report

Dear Authors!
The article seems appropriate for a second submission and reading. Thank you for considering the suggestions. The placement of the figures, the standardization of the format, and the significant increase in the number of references have benefited the article. 
I believe that all articles and analyses and works can be improved, but this one turned out quite well. I hope that research on this topic will continue and that further great analyses, and possibly a comprehensive new method, will be presented to standardise measurement indicators worldwide. Of course, you might also point out that a heterogeneous world is better...

Before the final submission, please review the article and resolve all references in the first place, e.g. IMDA, ARCEP, SMS, HD... If possible.

Best,

Author Response

Dear Reviewer,

We are grateful for your quick and astute comment. As suggested, we have defined the references as they first appear. Furthermore, typo, grammar, and punctuation errors are also modified. 

Best regards,